# Drug–Drug Interactions and Pharmacogenomic Evaluation in Colorectal Cancer Patients: The New Drug-PIN^®^ System Comprehensive Approach

**DOI:** 10.3390/ph14010067

**Published:** 2021-01-15

**Authors:** Michela Roberto, Alessandro Rossi, Martina Panebianco, Leda Marina Pomes, Giulia Arrivi, Debora Ierinò, Maurizio Simmaco, Paolo Marchetti, Federica Mazzuca

**Affiliations:** 1Oncology Unit, Department of Clinical and Molecular Medicine, Sant’Andrea Hospital, University “La Sapienza”, 00187 Rome, Italy; michela.roberto@uniroma1.it (M.R.); alessandro.rossi@uniroma1.it (A.R.); giulia.arrivi@uniroma1.it (G.A.); debora.ierino@uniroma1.it (D.I.); paolo.marchetti@uniroma1.it (P.M.); federica.mazzuca@uniroma1.it (F.M.); 2Department of Medical-Surgical Sciences and Translation Medicine, Sapienza University, Sant’Andrea University Hospital, 00187 Rome, Italy; 3Department of Neuroscience, Mental Health, 00187 Rome, Italy; ledama@hotmail.it (L.M.P.); maurizio.simmaco@uniroma1.it (M.S.); 4and Sensory Organs (NESMOS), Faculty of Medicine and Psychology, Sapienza University, Sant’Andrea University Hospital, 00187 Rome, Italy; 5Department of Advanced Molecular Diagnostics, Sant’Andrea Hospital, University “La Sapienza”, 00187 Rome, Italy

**Keywords:** advanced colorectal cancer, pharmacogenomic, drug–drug–gene interaction, Drug-PIN^®^

## Abstract

Drug–drug interactions (DDIs) can affect both treatment efficacy and toxicity. We used Drug-PIN^®^ (Personalized Interactions Network) software in colorectal cancer (CRC) patients to evaluate drug–drug–gene interactions (DDGIs), defined as the combination of DDIs and individual genetic polymorphisms. Inclusion criteria were: (i) stage II-IV CRC; (ii) ECOG PS (Performance status sec. Eastern coperative oncology group) ≤2; (iii) ≥5 concomitant drugs; and (iv) adequate renal, hepatic, and bone marrow function. The Drug-PIN^®^ system analyzes interactions between active and/or pro-drug forms by integrating biochemical, demographic, and genomic data from 110 SNPs. We selected DDI, DrugPin1, and DrugPin2 scores, resulting from concomitant medication interactions, concomitant medications, and SNP profiles, and DrugPin1 added to chemotherapy drugs, respectively. Thirty-four patients, taking a median of seven concomitant medications, were included. The median DrugPin1 and DrugPin2 scores were 42.6 and 77.7, respectively. In 13 patients, the DrugPin2 score was two-fold higher than the DrugPin1 score, with 7 (54%) of these patients experiencing severe toxicity that required hospitalization. On chi-squared testing for any toxicity, a doubled DrugPin2 score (*p* = 0.001) was significantly related to G3–G4 toxicity. Drug-PIN^®^ software may prevent severe adverse events, decrease hospitalizations, and improve survival in cancer patients.

## 1. Introduction

Drug–drug interactions (DDIs), defined as the result of metabolic interference between two or more drugs, thereby affecting their efficacy and/or toxicity, is a theme of major concern in medical oncology—especially due to their potential effect(s) on clinical outcomes [1].

Aside from well-known anticancer drugs, patients often take other medications to treat different conditions, such as diabetes and hypertension, among others, which could lead to severe changes in pharmacokinetic (absorption, distribution, metabolism, elimination) and pharmacodynamic properties, making this an even more complex scenario [2,3,4]. This is particularly relevant in older patients because cancer and ageing are both causes of polypharmacy in this population [5,6,7].

Equally important is the knowledge of a patient’s pharmacogenomic profile, including all cytochromes of the P450 (CYPs) enzyme superfamily and other genes affecting those properties, such as P-glycoprotein, ATP-binding cassette transporters, as well as detoxifying and DNA-repair enzymes [8,9]. Individual germline sequence variations in these genes, known as single-nucleotide polymorphisms (SNPs), affect inter-individual response(s) to drugs and reflect both the efficacy and toxicity profiles reported by patients [10].

A major example reflecting the importance of SNPs in clinical practice concerns point mutations affecting dihydropyrimidine dehydrogenase (DPD; encoded by the *DPYD* gene) which is implicated in the complex pathway of fluoropyrimidine metabolism (Figure 1).

Thus, testing DPD activity is strongly recommended before starting fluoropyrimidine treatment [11,12]. However, as extensively described by Palmirotta et al., many other SNPs, such as those affecting genes involved in irinotecan, oxaliplatin, or glutathione metabolism, could determine different responses and toxicities related to chemotherapy in gastrointestinal (GI) cancer patients [13].

In addition to pharmacogenetics, biochemical assays for pretreatment evaluation, such as an ex-vivo assays to determine the rate of peripheral blood mononuclear cell metabolism of 5-FU, known as the individual 5-FU degradation rate (5-FUDR), have also been developed and are suggested to be better and easy-to-use predictive factors of toxicity [14,15,16,17,18,19,20].

The combination of DDIs and individual genetic polymorphisms has recently contributed to the definition of a fascinating new concept known as drug–drug–gene interaction (DDGI), which may lead personalized medicine to another and more complicated level [21]. In fact, drugs often undergo multiple metabolic pathways and modifications that could affect plasma drug concentrations [22], especially when drug enhancers or inhibitors insist on a genetic predisposition [23]. To our knowledge, although DDI software is available as a tool that can be used in clinical practice [24], a system able to analyze and integrate information from both DDIs and individual genetic polymorphisms, and that could predict the impact of such DDGIs on patient outcomes, is still lacking.

Moreover, patients are still not routinely checked for both DDIs of concomitant medications and pharmacogenomic profile aside from DPD testing. This could lead to unexplained and serious adverse events (SAEs) in clinical practice during chemotherapy.

Herein, we discuss the use of Drug-PIN^®^, the first software that combines and simultaneously analyses the metabolic data, DDIs, and genomic profile of individual colorectal cancer patients undergoing 5-FU-based chemotherapy in an attempt to fill the gap of an important unmet need—a highly tailored approach for every single cancer patient. 

Thus, the objective of this study was to investigate the role of Drug-PIN^®^ software in the management of colorectal cancer patients and to correlate Drug-PIN^®^ results with the occurrence of toxicities.

## 2. Results

We screened 150 colon cancer patients who attended their initial oncological visit at our center. A total of 34 patients, who were candidates for 5-FU-based chemotherapy with multiple comorbidities for which they were taking at least five concomitant medications, were included in this study (Figure 2). Nineteen of thirty-four patients had advanced disease at diagnosis. A *KRAS* and/or *NRAS* mutation was identified in 14 patients, whereas in one case, a *V600E BRAF* mutation was identified. Their clinicopathological characteristics are reported in Table 1.

All patients submitted a blood sample for pharmacogenomic analysis and their data, concomitant therapies, and chemotherapy drugs were inputted to the Drug-PIN^®^ platform. Overall, the median DrugPin1 and DrugPin2 scores were 42.6 (range 11.1–180.3) and 77.7 (range 16.8–199.8), respectively. In 13 (38%) patients, the DrugPin2 score, which is the global score as described in the Methods section, was two-fold higher than the DrugPin1 score.

All patients were assessed for safety. The incidence of severe (G3–G4) hematological and gastrointestinal toxicity was 7 (20%) and 5 (15%), respectively. No severe hand-foot syndrome occurred. Seven patients experienced SAEs that required hospitalization: three for cardiotoxicity; two for profuse diarrhea; and the others for allergic reaction to oxaliplatin. In all these cases, when compared to DrugPin1, the DrugPin2 score was at least two-fold higher. All SAEs were completely addressed after hospital admission and no deaths due to toxicity occurred. On χ^2^ testing for severe toxicity due to any cause, a doubled DrugPin2 score (*p* < 0.0001) was shown to be significantly related to G3–G4 toxicity (Table 2).

Treatment was prematurely discontinued in 10 (56%) patients due to the occurrence of toxicity. However, in eight cases, chemotherapy was restarted using a 25% dose reduction with a favorable outcome, while two patients discontinued treatment entirely. 

With a median follow-up of 21 months, seven disease-related progressions and two deaths occurred.

## 3. Discussion

To the best of our knowledge, Drug-PIN^®^ is the first intelligent system to combine data regarding concomitant medications and a patient’s pharmacogenomic profile, DDIs, and metabolic data, resulting in a global score that can be used by the physician to better predict clinical outcomes and eventually prevent severe toxicities related to chemotherapy, to support any choices of therapeutic variation in the light of known interferences between drugs used in the therapy or unfavorable biochemical phenotypes.

In contrast to previous studies investigating pharmacodynamic, pharmacokinetic [2,5,6], and pharmacogenomic features [21,23,25], or involving other DDI software [24], we did not base our analysis solely on DDI aspects [1,2,5,6], link a single SNP with a certain drug toxicity [12,26,27,28,29,30,31,32,33], or study a single drug-related toxicity [16,17,18,19,20].

Our Drug-PIN^®^ platform gathers all of the patient’s traits, including gender, age, weight, lifestyle habits, concomitant medications, and pharmacogenomics; thus, we are able to summarize all of these aspects in a single global score. Therefore, the Drug-PIN^®^ score represents the patient’s pharmacological profile globally, with their unique features collectively considered.

In fact, we defined the following crucial scores for each patient: DDI, DrugPin1, and DrugPin2. The DDI score is derived from the interaction(s) between medication(s) taken by a patient. Basically, this is the “first step” of a complete evaluation. The DrugPin1 score integrates medications with the genomic profile of each individual, and reveals a complex and unique drug–patient interaction that could lead to unexpected favorable or unfavorable outcomes. DrugPin2 is the “last step”, adding chemotherapy to DrugPin1 in order to highlight potentially dangerous interactions between the patient and the specific oncological treatment. Each of these steps must be monitored and considered individually in order to better understand where potentially relevant interactions may occur and provide a prompt solution. Most of the currently available software does not integrate DDI with such a wide range of single-nucleotide polymorphisms and, moreover, do not suggest which is the best available treatment for a particular patient, relying on their individual profile. This is a completely new approach that takes personalized medicine to a new level.

In fact, as reported in the univariate analysis for severe toxicity (Table 2), the mean DDI score was not significantly different between patients who developed severe toxicity and those who did not. Similarly, no significant difference was reported between DrugPin1 and DrugPin2 scores considered as absolute value, although some patients showed high—potentially dangerous—scores. It is possible that patients susceptible to DDIs according to their Drug-PIN^®^ scores did not develop any severe toxicity thanks to compensatory enzymatic mechanisms, which may have “cushioned” any resulting change in drug metabolism [34].

However, in 13 (38%) patients, the DrugPin2 score was considerably different from the DrugPin1 score, and thus they experienced a significantly higher incidence of G3–G4 toxicities (91% vs. 9%, *p* < 0.0001) (Table 2). No other clinicopathological characteristics were found to be related to the occurrence of severe toxicities, even if all patients treated with a FOLFIRI regimen developed G3–G4 adverse events. A previous study reported a higher toxicity of FOLFOX when compared with the FOLFIRI regimen in terms of hematological, gastrointestinal, and neuropathic toxicity [35,36]. However, the two patients treated with FOLFIRI that experienced G3–G4 toxicities in our study were homozygous and heterozygous for uridine diphosphate glucuronosyltransferase (UGT) 1A1*28 polymorphism, respectively. In fact, polymorphisms in the UGT1 enzyme subfamily, especially the *UGT1A1* gene, can dramatically affect clinical outcomes and drug-related toxicities [30]. Based on these findings, the FDA and other regulatory agencies recommend a dose reduction in patients homozygous for UGT1A1*28, while no dose modification is suggested for those with UGT1A1*28 heterozygosity [37].

Our study certainly had some limitations that should be considered. This is a brief report describing a platform application with a limited number of patients. As such, the results obtained, although interesting, need to be validated in larger cohorts. Furthermore, there was only a short-term follow-up, and data regarding disease-free, progression-free, and overall survival are still immature.

However, this is the first example of software capable of combining all patient features in a score that the clinician can easily use as a predictive marker of toxicity before starting chemotherapy. In addition, this tool can be used for interdisciplinary consultation because it could provide easy-to-use suggestions such as “usable”, “non-preferential”, and “not recommended” for each drug. The program offers possible therapeutic alternatives, or suggests drugs that support the main therapy (e.g., common pain therapy) in a multidisciplinary context, supporting the choice of molecules that are the most effective in therapeutic terms and lead to less interference with each drug and with the main specific oncological therapy.

Therefore, the Drug-PIN^®^ program is a supportive technology in the choice of drug(s). Adverse reactions related to drug–drug–gene interactions which are considered by the program relate to much of the well-established and well-known literature. In addition, the program updates its data on drug–drug interactions, and any changes in efficiency and/or toxicity related to biochemical phenotypes of one or more polymorphisms. All of this information is transformed into a numerical figure using a copyrighted algorithm, so as to calculate a score that returns an overview quickly and without neglecting important information for the purposes of therapy guiding the same according to the cardinal principle of personalized medicine of “primum non nocere”.

Thus, Drug-PIN^®^ enables physicians to fine-tune patient’s therapy by simulating various drug combinations and assessing adverse drug interactions. Finally, it produces a global report that can be shared with the patient and used to support medical professionals in deciding on the most effective and low-risk pharmaceutical prescriptions. In order to deliver a personalized report suggesting the best medication solutions for the patient, Drug-PIN^®^ supports them with a modern technology that analyses unique patient data with official medical data, therefore improving clinical outcomes and reducing negative drug reactions. 

The Drug-PIN^®^ system technology cannot predict a specific toxicity that will affect a certain organ or site, but can usefully predict which patients are more prone to develop toxicities according to their pharmacogenomic profile and medical history of concomitant medications. For example, the central and peripheral nervous systems can be affected by 5-FU treatment, with nerve palsy a feared complication [38,39,40]. This is more relevant when 5-FU is combined with other drugs, especially cisplatin [41]. Currently available 5-FU metabolism pre-treatment assays, together with DYPD, MTHF, and other genes’ genomic status could help the clinician to avoid such toxicities by reducing the dose or selecting a different drug for the patient.

## 4. Materials and Methods

Drug-PIN (Personalized Interactions Network) is the first intelligent system that recognizes the critical role of multiple interactions between active and/or pro-drug forms by integrating data related to the following parameters: (i) the functional biochemical profile of each patient; (ii) patient age; (iii) biochemical indicators of both renal (creatinine and calculated GFR) and liver function (ALT and AST); (iv) lifestyle habits such as the consumption of alcohol, coffee, and/or cigarettes; and (v) ongoing therapeutic drug therapy. The Drug-PIN^®^ technology can best be described as “functional biochemistry based on genomic profile” (http://www.drug-pin.com/index.html). Developed by university researchers over the past decade, Drug-PIN^®^ is a multidimensional, comprehensive approach to personalized drug therapies. Moreover, owing to its easy-to-use technology, it aims to support physicians in deciding on the best prescription choice and improving patient clinical outcomes, as well as by preventing potentially dangerous DDIs. 

The functional biochemistry profile is based on gene mutations affecting selected crucial SNPs. In fact, these SNPs have been described in several scientific reports, which have demonstrated a relationship with functional metabolic changes in enzymes, receptors, and transporters, thus affecting both pharmacokinetic and pharmacodynamic aspects.

The Drug-PIN^®^ system takes into consideration the genomic structure of 110 SNPs for genes coding for Phase 1 enzymes, Phase 2 enzymes, receptors, and transporters. These SNPs are used to create a panel for targeted DNA re-sequencing (Appendix A).

Patient DNA is extracted from 5 mL samples of peripheral blood using the automatic QIA Symphony system for the extraction of nucleic acids (Qiagen, Hilden, Germany); the latter is then processed using a next-generation sequencing platform, the Ion Chef/Ion S5 System (Thermo Fisher Scientific, Waltham, MA, USA), according to the manufacturer’s instructions.

The results of DDGI analysis performed by the Drug-PIN^®^ software are represented by a numerical score to be understood in a penalizing sense: the greater the associated number, the more dangerous the drug cocktail will be. This numerical score is derived by considering the following four different portions: the first portion is the inherent properties of the DDIs; the second portion is linked to the patient’s age and the drug cocktail, identifying any inadequacy on the basis of the Priscus list [42]; the third portion is related to renal function which, similarly to the previous one, affects the score following the indications of appropriateness dictated by the Beers list [43]; and the final portion is related to drug metabolism, calculated on the basis of any pharmacokinetic problems expected in association with the analyzed SNPs.

In this way, the software supports physicians in considering more targeted therapeutic choices and enables them to build, according to the specific needs of the patient, new therapeutic combinations. The system is able, in fact, to automatically evaluate and recalculate the score every time a new drug is added, with the aim of suggesting the best therapeutic options for each class of drugs. Drug-PIN^®^ uses a base of official data from the European Agency of Medicines, FDA, CPIC/Pharm GKB, and Open Data.

Patient DNA is extracted from 5 mL samples of peripheral blood, using the automatic QIA Symphony system for the extraction of nucleic acid (Qiagen, Hilden, Germany) using the dedicated kits produced by the manufacturer of the instrument; the latter is then processed using a Next-Generation Sequencing platform, Ion Chef/Ion S5 system (Thermo Fisher Scientific, Waltham, MA, USA), using the dedicated kits produced by the manufacturer of the instrument, according to the manufacturer’s instructions.

In this study, we defined the following crucial scores for each patient: (i) DDI is the score between concomitant medications the patient is taking; (ii) DrugPin1 is the score calculated before chemotherapy, including concomitant medications and SNPs profile; and (iii) DrugPin2 is the global score, including the DrugPin1 and chemo drugs in the software analysis. 

### 4.1. Patients

This study prospectively enrolled GI cancer patients who underwent multidisciplinary team treatment from January 2018 at our center. Inclusion criteria included: (i) histologically confirmed diagnosis of colorectal cancer (stage II–IV); (ii) age ≥ 18 years; (iii) ECOG performance status ≤ 2; (iv) taking at least 5 drugs for comorbidities, by definition of polypharmacy [5,6,44,45]; (v) no previous chemotherapy treatment; and (vi) adequate renal, hepatic, and bone marrow function. Meanwhile, the presence of one of the following criteria excluded a patient’s participation in the study: GI tumors that had received any standard treatment(s), cardiovascular or central nervous system (CNS) disease, previously untreated CNS metastases, pregnant or breast-feeding patients, organ dysfunction that usually contraindicate the use of cytotoxic drugs, and/or substance abuse and any other psychological condition that may interfere with the evaluation of study results. Patients with concomitant infectious diseases were also excluded from enrollment. All patients were candidates for 5-FU-based chemotherapy (allowed protocols: capecitabine 1000 mg/m^2^ twice daily (day 1, evening, to day 15, morning), every 3 weeks; FOLFOX6, oxaliplatin 100 mg/m^2^ IV infusion, given as a 120 min IV infusion in 500 mL D5W, concurrent with levoleucovorin 200 mg/m^2^ IV infusion, followed by 5-fluorouracil 400 mg/m^2^ IV bolus, followed by 46 h 5-FU infusion (2400 mg/m^2^); XELOX, intravenous oxaliplatin 130 mg/m^2^ (day 1) followed by oral capecitabine 1000 mg/m^2^ twice daily (day 1, evening, to day 15, morning), every 3 weeks; FOLFIRI, irinotecan 180 mg/m^2^ IV infusion, administered as a 120 min IV infusion in 500 mL D5W, concurrent with levoleucovorin 200 mg/m^2^ IV infusion, followed by 5-FU 400 mg/m^2^ IV bolus, followed by 46-h 5-FU infusion (2400 mg/m^2^), eventually combined with the anti-VEGF, bevacizumab, or anti EGFR targets, according to the tumor molecular profile. A blood sample for DNA extraction and analysis was obtained at the beginning of treatment. Demographic information, medical history, and adverse drug reaction data were collected. Dose reductions in chemotherapy, as well as treatment delays and permanent discontinuations, were also recorded. All patients provided written informed consent prior to blood withdrawal and pharmacogenetic analysis. In each patient, toxicity was regularly assessed and graded according to Common Terminology Criteria for Adverse Events (CTCAE) v4.03. Toxicity was defined as absent or mild when the severity grade was between 0 and 2, and as severe when the severity grade was 3 or 4. The study was conducted in accordance with the Declaration of Helsinki, and all patients signed informed consent for scientific research purposes at the first oncological visit. Due to the observational, non-interventional nature of this study, notification was sent to the local Ethics Committee (Comitato Etico Sapienza Università di Roma, Rome, Italy) (normative ref. GU della Repubblica Italiana n.76 of 31 March 2008).

### 4.2. Statistical Analysis

According to the aforementioned background, in order to provide an adequate sample size, assuming a 30% moderate–severe toxicity with 5-FU-based chemotherapy, a total of 81 patients were enrolled (calculated with an error of 10% and confidence interval (CI) of 95%). However, a feasibility study on the first cases enrolled is described in this publication. Associations between severe toxicities and clinicopathological characteristics, chemotherapy regimens, and Drug-PIN^®^ scores were assessed using the χ^2^ test and *t*-test to compare frequencies and means, respectively. A *p*-value < 0.05 was considered to be statistically significant. SPSS version 24 (SPSS Inc. Chicago, IL, USA) was used.

## 5. Conclusions

This study highlights the promising use of Drug-PIN^®^ software in colon cancer patients, which aims to enhance the efforts of personalized medicine by tailoring therapies to individual patients in order to prevent toxicity, increase compliance, and, therefore, survival. The same system is configured as a valid instrument of important support in the context of other medical specialties as well, and it could be the starting point for the choice of chronic therapies free from possible known unfavorable interactions, within the limits of those possible that have been reported in the literature. Given these encouraging results, further studies are needed to confirm our data and better investigate the utility of Drug-PIN^®^ in the new era of personalized medicine with remarkable effect and functionality in cancer patients. 

## Figures and Tables

**Figure 1 pharmaceuticals-14-00067-f001:**
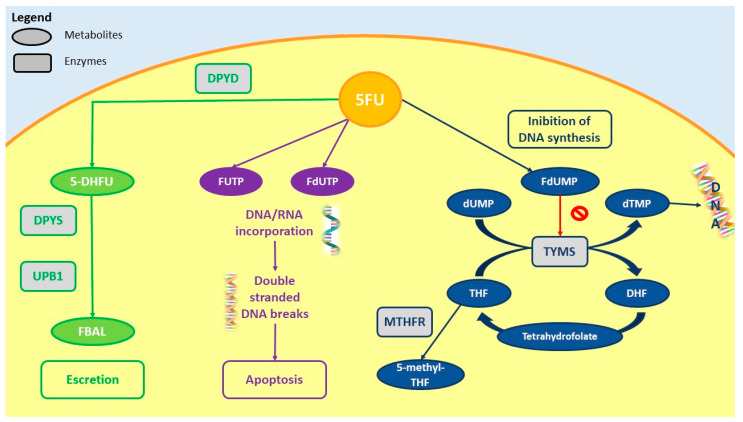
Metabolism and mechanism of action of 5-fluorouracil (5-FU) in colon cancer tissue. Liver dihydropyrimidine dehydrogenase (DPD) is responsible for about 90% of fluoropyrimidine metabolism by converting them into 5-DHFU. TYMS is an enzyme that plays a crucial role in the early steps of DNA biosynthesis by converting dUMP to dTMP using folate, donated by MTHFR and then incorporated in DNA and RNA as pyrimidine analogues, with consequent apoptosis in cancer cells. MTHFR usually converts THF in 5-methyl THF and uses it as a methyl donor. dUMP forms a ternary complex with THF and TYMS, with inhibition of the enzyme; thus, MTHFR deficiency could be responsible for 5,10-methylene-THF increase in cells, enhancing the inhibitory complex and fluoropyrimidine toxicity. TYMS polymorphisms could also be responsible for 5-FU-related toxicities. However, the roles of both MTHFR and TYMS polymorphisms still need to be translated to clinical practice, as the results of in vivo studies to date have been controversial; as such, there is no indication for testing them to drive clinical decisions. In contrast, testing DPD activity is strongly recommended before starting fluoropyrimidine therapy. In fact, DPD deficiency significantly increases the risks for the development of severe 5-FU-associated toxicity.

**Figure 2 pharmaceuticals-14-00067-f002:**
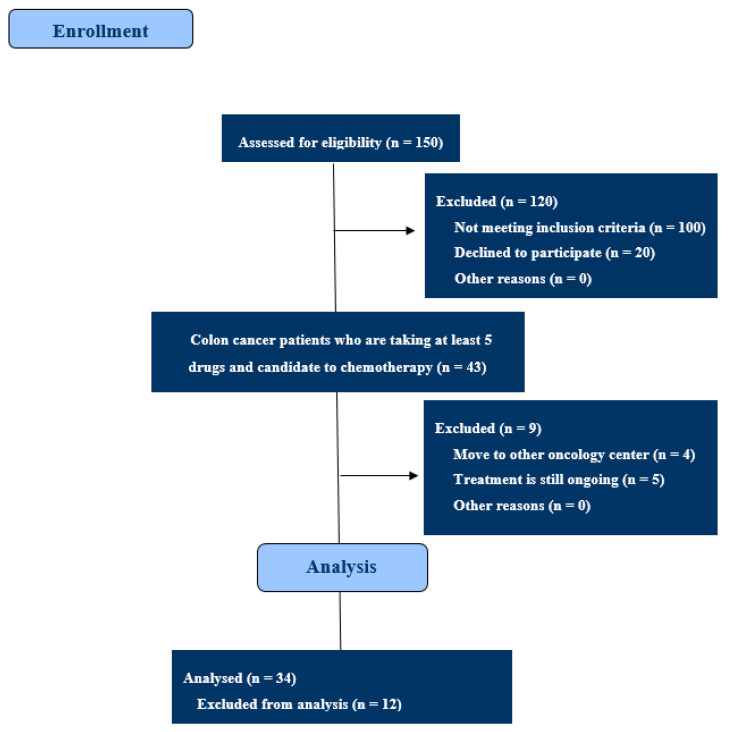
CONSORT diagram. One hundred and fifty colon cancer patients were assessed for eligibility, of whom 120 were excluded: 100 patients assumed <5 concomitant drugs, and 20 declined to participate. There were 43 colon cancer patients who were taking at least 5 drugs and were candidates for chemotherapy, of whom 12 were excluded from analysis.

**Table 1 pharmaceuticals-14-00067-t001:** Clinicopathologic features (valid cases and percentages).

**Total**	34	100
**Median Age (range)**	73 (55–83)	
**Gender**		
Male	23	68
Female	11	32
**Charlson Comorbidity Index**		
≤8	16	47
>8	18	53
**N. Concomitant Medications**		
Median (range)	7 (5–9)	
**Drug Class**		
Antidiabetic	12	53
Proton-pump inhibitor	24	71
Antihypertensive	27	79
Antihyperlipidemic	16	47
Antiarrhythmic	18	53
Antithrombotic	28	82
Corticosteroids	4	12
Thyroid hormone replacement	5	15
**Stage of Disease at Diagnosis**		
II–III	17	50
IV	17	50
**ECOG Performance Status**		
0	18	53
≥1	16	47
**Chemotherapy Regimen**		
FOLFOX/XELOX	24	71
FOLFIRI	2	6
Capecitabine	8	23

**Table 2 pharmaceuticals-14-00067-t002:** Correlation between any toxicity and clinical characteristics of the study patients.

Factor N (%)	Any Toxicity		*p*	HR (95% CI)
	G0–G2	G3–G4		
**Age, years**			0.661	0.70 (0.1–3.4)
≤75	15 (65)	8 (73)		
>75	8 (35)	3 (27)		
**Gender**			0.259	2.36 (0.5–10.7)
Male	17 (74)	6 (55)		
Female	6 (26)	5 (45)		
**Stage**			0.271	0.44 (0.1–1.9)
II-III	10 (43)	7 (64)		
IV	13 (57)	4 (36)		
**ECOG PS**			0.897	0.90 (0.2–3.8)
0	12 (52)	6 (54)		
≥1	11 (48)	5 (45)		
**Chemotherapy**			0.06	7.05 (0.8–57.8)
FOLFOX/XELOX	16 (70)	8 (73)		
FOLFIRI	0 (0)	2 (18)		
Capecitabine	7 (30)	1 (9)		
**Charlson Comorbidity Index**			0.545	0.64 (0.1–2.7)
≤8	10 (43)	6 (54)		
>8	13 (57)	5 (45)		
**N. Concomitant Medications**			0.538	1.62 (0.3–7.5)
≤7	17 (74)	7 (64)		
>7	6 (26)	4 (36)		
**Drug Class**				
Antidiabetic	10 (43)	2 (18)	0.149	0.29 (0.1–1.6)
Antihypertensive	18 (78)	9 (81)	0.81	1.25 (0.2–7.7)
Proton-pump inhibitor	15 (65)	9 (82)	0.32	2.4 (0.4–13.9)
Corticosteroids	3 (13)	1 (9)	0.738	0.66 (0.1–7.2)
Antilipidemic	9 (39)	8 (64)	0.18	2.72 (0.6–12.0)
Antiarrhythmic	13 (56)	5 (45)	0.545	0.64 (0.1–2.7)
Anticoagulant	20 (87)	8 (73)	0.309	0.40 (0.1–2.4)
Thyroid hormone replacement	2 (9)	3 (27)	0.152	3.93 (0.5–28.1)
**DDI score**				
Mean ± SD	24 ± 13	25 ± 14	0.853	
**DrugPin1**				
Mean ± SD	61 ± 48	36 ± 29	0.271	
**DrugPin2**				
Mean ± SD	85 ± 52	96 ± 41	0.687	
**DrugPin2 > DrugPin1**			<0.0001	66.6 (6.1–725.4)
<50%	20 (87)	1 (9)		
≥50%	3 (13)	10 (91)		

## Data Availability

Data are contained within the article and Appendix A.

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
