# Peer review of "Drug–Drug Interactions and Pharmacogenomic Evaluation in Colorectal Cancer Patients: The New Drug-PIN® System Comprehensive Approach"

_pharmaceuticals, 2021, doi:10.3390/ph14010067_

Round 1
Reviewer 1 Report
In this manuscript, the authors suggested that Drug-Drug Interaction (DDI) may affect both treatment efficacy and toxicity. In general, the manuscript is well written and the contents support their conclusion. According to the data presented, their claim seems to be sound. The data brought by this article are valuable because of the relative paucity of information available on that topic. This study seems very interesting. However, I have few concerns before making this paper fully acceptable for publication in pharmaceutics.
Minor concerns:
- Abbreviations must be included in the beginning of the manuscript especially after abstract; otherwise it becomes very difficult to read this manuscript.
- The objective ot the study is not clear
- The line 63-69 should be placed after abstract
- Figure 1 and Figure 2 legends are not clear and images are of poor quality
- The power analysis G sore data is not providing for sample size. Power analysis must be performed in clinical studies is of paramount importance.
- Under what criteria this study was approved by ethical committee? Exclusion criteria must be mentioned.
Major concerns:
- Majority of patients are resistant to 5-FU based chemotherapy in colon cancer . Due to Kras and BRAF mutation .Why did not authors included such tumor molecular profile in this study ? as there tumor molecular profile make 5-FU resistant in colorectral cancer .
- 5-Fu based chemotherapy is also having severe secondary complications associated with it such as craninal (oculomotor ) nerve plasy. Can authors explain how these issues will be resolved by using Drug-pin system technology?
- One of the difficulty of pharmacogenomics studies are replication of similar results and rare adverse drug reactions and heterogensity of drug response phenotype .Can authors explains how these complications will be overcome by using their technology?
- Proper assessment and statistical analysis is required .
- Follow of study with larger sample size are required and also survival cures by employing Kaplan –meier method are of significant importance .
- Comparison of different groups of patients using lonf –rank test is missing?
- The cox propotional hazard model must be included for multiviariate analysis of the prognostic factors such as the patient, tumor and treatment modalities ?.
- Discussion of the manuscript is not upto the mark need tailoring especially in conclusion as there is more research needed to fully understand the role of this technology by analysing all the pros and cons .
Overall, Drug-pin system technology is innovative and the strength of this manuscript. However, major weaknesses such as experimental design, lack of focus and organization of the manuscript lessens my enthusiasm.
Author Response
We are very pleased to receive your in-depth review of our manuscript. We appreciate your time and efforts in helping us improve our paper. We read your comments carefully and made corrections and revisions wherever possible. Our responses to each comment are below. We are looking forward to your further recommendations.
Minor concerns:
- Abbreviations must be included in the beginning of the manuscript especially after abstract; otherwise it becomes very difficult to read this manuscript.
Answer to “Minor concerns” question 1. Thanks for your suggestion, now we have added after abstract the following abbreviations in Pages 1-2, Lines 34-44: “Abbreviation: DDI, Drug-Drug Interaction; CRC, colorectal cancer; DDGI, Drug-Drug-Gene Interaction; CYPs, cytochromes P450 enzyme; SNPs, single nucleotide polymorphisms; 5FU, 5-Fluorouracil; FdUMP, 5-fluoro-2′-deoxyuridine-5′-monophosphate; TYMS, thymidylate synthase; dUMP, deoxyuridine monophosphate; dTMP, deoxythymidine monophosphate; THF, 5,10 metylene tetrahydrofolate; DHF, 7, 8 dihydrofolate; 5-methyl-THF, 5 methyl tetrahydrofolate; MTHFR methylenetetrahydrofolate reductase; FdUTP, fluorodeoxyuridine triphosphate; FUTP, fluorouridine triphosphate; DYPD, dihydropyrmidinase; 5-DHFU, 5-dihydrofluorouracil; DPYS, dihydropyrmidinase; UPB1, beta-ureidopropionase; FBAL, fluoro-beta-alanine; GI, gastrointestinal; 5-FUDR , 5-FU degradation rate; SAE, serious adverse events; CTCAE, Common Terminology Criteria for Adverse Event; UGT, uridine diphosphate glucuronosyltransferase; DFS, disease-free survival; PFS, progression-free survival; OS, overall survival”.
- The objective of the study is not clear
Answer to “Minor concerns” question 2. According to exploratory nature of this study we have not pre-planned any primary end-points. However now we have clarified our objective at the last of the introduction in Page 4, Lines 128-129: “we discuss the use of Drug-PIN®, the first software that combines and analyses simultaneously metabolic data, DDI and genomic profile of each individual, in colorectal cancer patients receiving 5-FU based chemotherapy, to explore its potential utility in clinical practice. Thus, the objective of this study is to investigate the role of Drug-PIN® software in colorectal cancer patients management and to correlate Drug-PIN® results with toxicities occurrence.”
- The line 63-69 should be placed after abstract.
Answer to “Minor concerns” question 3. According to your suggestion, the line 63-69 have been placed after the abstract, in the paragraph “Abbreviations” in Pages 1-2, Lines 34-44.
- Figure 1 and Figure 2 legends are not clear and images are of poor quality.
Answer to “Minor concerns” question 4. According to your suggestion, we have modified the legend of Figure 1 in Page 3, Lines 75-87 as follow: “Liver DPD is responsible for about 90% of fluoropyrimidines metabolism, by converting them into 5-DHFU. TYMS is an enzyme that plays a crucial role in early steps of DNA biosynthesis by converting dUMP to dTMP using folate, donated by MTHFR and then incorporated in DNA and RNA as pyrimidines analogues, with consequent apoptosis in cancer cells. MTHFR usually converts THF in 5-methyl THF and uses it as methyl donor. dUMP forms a ternary complex with THF and TYMS, with inhibition of the enzyme; thus, MTHFR deficiency could be responsible for 5,10 metylene THF increase in cells, enhancing the inhibitory complex and fluoropirimidines toxicity. TYMS polymorphisms, as well, could be responsible for 5FU-related toxicities. However, roles of both MTHFR and TYMS polymorphisms still need to be translated in clinical practice, as the results of in vivo studies have been controversial so far; to date, there is no indication for testing them to drive clinical decisions. On the contrary, testing DPD activity is strongly recommended before start fluoropyrimidines. In fact, DPD deficiency significantly increased risks for developing a severe 5FU-associated toxicity”.
Unfortunately it was not possible to further increase the quality of Figure 1, but we are willing to send its original format in “Power Point” to have a higher display quality.
We have replaced Figure 2 with an image of more quality, changing the colours of the boxes (blue) and words (white), thus making more visible the content, and we have added a legend for Figure 2 in Page 8, Lines 276-279, as follow reported: “150 colon cancer patients were assessed for eligibility. 120 out of 150 cases were excluded: 100 patients, assumed <5 concomitant drugs and 20 declined to participate. Colon cancer patients who were taking at least 5 drugs and candidate for chemotherapy were 43. Of these, 12 were excluded from analysis”.
- The power analysis G sore data is not providing for sample size. Power analysis must be performed in clinical studies is of paramount importance.
Answer to “Minor concerns” question 5. We have added in Page 6, Lines 215-219: “According to the aforementioned background, in order to provide an adequate sample size, assuming a 30% of moderate-severe toxicity with 5-FU based chemotherapy , a total of 81 patients will be enrolled (calculated with an error of 10% and confidence interval (CI) of 95%). However, a feasibility study analysis on the first cases enrolled was described in this publication.”
- Under what criteria this study was approved by ethical committee? Exclusion criteria must be mentioned.
Answer to “Minor concerns” question 6. We have reported patient’s criteria in material and method section, and now we have added in Page 5 , Lines 184-190 the following text “Meanwhile, the presence of one of the following criteria excluded patients’ participation to the study: GI tumours that has had received any standard treatments, cardiovascular or central nervous system (CNS) disease, previously untreated CNS metastases, pregnant or breast-feeding patients, organ dysfunctions that usually hinder the use of cytotoxic drugs, and/or substance abuse and any other psychological condition that may interfere with the evaluation of study results. Patients with concomitant infectious diseases were also excluded from the enrollment”.
Major concerns:
- Majority of patients are resistant to 5-FU based chemotherapy in colon cancer . Due to Kras and BRAF mutation .Why did not authors included such tumor molecular profile in this study ? as there tumor molecular profile make 5-FU resistant in colorectral cancer.
Answer to “Major concerns” question 2: we have added se following data in Pages 6, Lines 227-229: “Nineteen out of 34 patients had an advanced disease at diagnosis. A KRAS and/or NRAS mutation was identified in 14 patients, whereas in one case was identified the V600E BRAF mutation”.
- 5-Fu based chemotherapy is also having severe secondary complications associated with it such as craninal (oculomotor ) nerve plasy. Can authors explain how these issues will be resolved by using Drug-pin system technology?
Answer to “Major concerns” question 2: Thanks for your suggestion, we have added the following text in Page 12, Lines 383-390: “Drug-PIN® system technology cannot predict a specific toxicity that will affect a certain organ or site, but can usefully predict which patients are more prone in developing toxicities, according to their pharmacogenomic profile and medical history of concomitant medications. For example, central and peripheral nervous systems can be affected by 5-FU treatment, with nerve palsy as a fearful complication [42–44]. This is more relevant when 5FU is combined with other drugs, especially cisplatin [45]. The 5-FU metabolism pre-treatment assays, together with DYPD, MTHF and other genes genomic status could help the clinician avoiding such toxicities, by reducing the dose or selecting a different drug for the patient”.
- One of the difficulty of pharmacogenomics studies are replication of similar results and rare adverse drug reactions and heterogeneity of drug response phenotype . Can authors explains how these complications will be overcome by using their technology?
Answer to “Major concerns” question 3. As your suggestion we have added in the discussion the following paragraph in Page 12, Lines 368-375: “Therefore, the Drug-PIN® program is a support technology in the choice of the drug. The adverse reactions, related to drug-drug-gene interaction and which are taken into account by the program, relate to much of the well-established and well-known literature. In addition, the program updates its data on drug-drug interactions, and any changes in efficiency and / or toxicity related to biochemical phenotypes of one or more polymorphisms. Each of this information is transformed into a numerical figure, using an algorithm copyrighted, so as to calculate a score that returns an overview quickly and without neglecting important information for the purposes of therapy guiding the same according to the cardinal principle of personalized medicine of "primum non nocere”.
- Proper assessment and statistical analysis is required.
Answer to “Major concerns” question 4. We have reported statistical analysis in Page 6, Lines 215-219 and modified the paragraph as your suggestion “According to the aforementioned background, in order to provide an adequate sample size, assuming a 30% of moderate-severe toxicity with 5-FU based chemotherapy, a total of 81 patients will be enrolled (calculated with an error of 10% and confidence interval (CI) of 95%). However, a feasibility study analysis on the first cases enrolled was described in this publication. Associations between severe toxicities and clinicopathological characteristics, chemotherapy regimens and Drug-PIN® scores were assessed using the χ2–test and t-test to compare frequencies and means, respectively. A P value < 0.05 was considered as statistically significant. SPSS statistical software, version 24 (SPSS Inc. Chicago, Illinois, USA) was used.”
- Follow of study with larger sample size are required and also survival cures by employing Kaplan –meier method are of significant importance .
Answer to “Major concerns” question 5. Thanks for your observation. This is an exploratory analysis on first patients who was evaluated in the new software DrugPin. We are conscious of limited sample size and that survival data are still immature. Indeed, we have reported in discussion section in Page 12, Lines 355-359, the limit of our study by the text “Our study certainly presents some limitations to be taken into account: this is a brief report describing platform application with a limited number of patients; obtained results, although interesting, need to be validated with larger cohorts. Furthermore, there is a short-term follow-up, and data regarding disease-free (DFS), progression-free (PFS) and overall (OS) survival are still immature”.
- Comparison of different groups of patients using lonf –rank test is missing?
Answer to “Major concerns” question 6. Since data of survival are still immature (only 21 months of follow up for early and later stage colorectal cancer is not enough to identify any survival difference), no log-rank test for survival was calculated.
- The cox propotional hazard model must be included for multiviariate analysis of the prognostic factors such as the patient, tumor and treatment modalities ?
Answer to “Major concerns” question 7. Multivariate analysis, since no factor with the except of DrugPin scores resulted significant at the univariate analysis, was not calculated. This first study highlights the promising approach of the Drug-PIN® software in colon cancer patients treatment, even if further studies are needed to confirm our data and better investigate the effect of Drug-PIN® in the new era of personalized medicine in cancer patients. However, given the encouraging results and the potential utility of this software has showed in this first experience, we would like to conduct a larger sample study with a longer follow up with the aim to present in the next studies survival curves and log-rank test analysis. We hope that your consideration for this preliminary, descriptive analysis was positive, considered the innovative technology of the software described in the present study.
- Discussion of the manuscript is not upto the mark need tailoring especially in conclusion as there is more research needed to fully understand the role of this technology by analysing all the pros and cons . ?
Answer to “Major concerns” question 8.
According to your suggestion we have added the following text to “Discussion”:
- In Page 11, Lines 313-315: “to support any choices of therapeutic variation in the light of known interferences between drugs in therapy or unfavourable biochemical phenotypes”;
- In Page 12, Lines 363-367: “The program offers possible therapeutic alternatives, or suggests drugs that support the main therapy (think of the common pain therapy) in a multidisciplinary context, supporting the choice of molecules that are the most effective in therapeutic terms and less interfering with each drug and with the main specific oncological therapy”.
According to your suggestion we have added the following paragraphs to “Conclusion”:
- In Page 12, Lines 396-399: “The same system is configured as a valid instrument of important support in the context of other medical specialties as well, it could be the starting point for the choice of chronic therapies free from possible known unfavorable interactions, within the limits of the possible demonstrated in the literature”.
- In Page 12, Lines 400-401: “with remarkable affection and functionality”.
Reviewer 2 Report
- A graphical abstract is missing
- Please ensure the last paragraph of the introduction highlighting the novelty of the approach and the gap that this research fills
- Figure legends should be more comprehensive and understood without the need of going back to the manuscript. Ensure that figure 1 highlight the key proses and label the abbreviated enzymes and process within the legend (FUtp, FdUTP, etc)
- From line 63-69, it is in the wrong position I think this should be part of the figure legend
- In vivo, vitro, ex vivo it should be written in italic please revise
- Is there a material section please list all chemical used and their suppliers
- Authors should provide an IRB reference for samples
- Figure 2, same comment for the legend, and please use a clearer and better-quality font it is distorted
- Could you please provide a better-quality Figure 2 very difficult to follow or use a yellow highlight with a black font?
- If possible, replace the word “sex” in Table 1 with Gender and comprehensive legend
- Please align the G0-G2, G3G4 in table 2, and replace “sex” with gender
- Please include the conclusion section Information, Author Contributions, Conflict of Interest and other Ethics Statements following the journal formatting
- In the discussion could the authors add their reasoning of the section of DDI, DrugPin1, and DrugPin2 scores, and how do the findings corollate with previously published data
- Please include key references
10.1038/s41397-019-0122-0
https://www.jpsmjournal.com/article/S0885-3924(19)30057-0/fulltext
https://doi.org/10.3389/fphar.2016.00071
Author Response
We are very pleased to receive your in-depth review of our manuscript. We appreciate your time and efforts in helping us improve our paper. We read your comments carefully and made corrections and revisions wherever possible. Our responses to each comment are below. We are looking forward to your further recommendations.
- A graphical abstract is missing
Answer to question 1: Thanks for your suggestion, we have elaborated and added a graphical abstract in Page 2, Line 53.
- Please ensure the last paragraph of the introduction highlighting the novelty of the approach and the gap that this research fills ?
Answer to question 2: Thanks for your suggestion, we have modified the last paragraph of the introduction, in Page 4, Lines 123-126, as follows: “Hereby we discuss the use of Drug-PIN®, the first software that combines and analyses simultaneously metabolic data, DDI and genomic profile of each individual, in colorectal cancer patients receiving 5-FU based chemotherapy, trying to fill the gap of an important unmet need: an highly tailored approach for every single cancer patient.”
- Figure legends should be more comprehensive and understood without the need of going back to the manuscript. Ensure that figure 1 highlight the key proses and label the abbreviated enzymes and process within the legend (FUtp, FdUTP, etc).
Answer to question 3: According to your suggestion, we have included all abbreviations used in the text in the beginning of the manuscript, in Pages 1-2 lines 34-44, as follow: “Abbreviation: DDI, Drug-Drug Interaction; CRC, colorectal cancer; DDGI, Drug-Drug-Gene Interaction; CYPs, cytochromes P450 enzyme; SNPs, single nucleotide polymorphisms; 5FU, 5-Fluorouracil; FdUMP, 5-fluoro-2′-deoxyuridine-5′-monophosphate; TYMS, thymidylate synthase; dUMP, deoxyuridine monophosphate; dTMP, deoxythymidine monophosphate; THF, 5,10 metylene tetrahydrofolate; DHF, 7, 8 dihydrofolate; 5-methyl-THF, 5 methyl tetrahydrofolate; MTHFR methylenetetrahydrofolate reductase; FdUTP, fluorodeoxyuridine triphosphate; FUTP, fluorouridine triphosphate; DYPD, dihydropyrmidinase; 5-DHFU, 5-dihydrofluorouracil; DPYS, dihydropyrmidinase; UPB1, beta-ureidopropionase; FBAL, fluoro-beta-alanine; GI, gastrointestinal; 5-FUDR , 5-FU degradation rate; SAE, serious adverse events; CTCAE, Common Terminology Criteria for Adverse Event; UGT, uridine diphosphate glucuronosyltransferase; DFS, disease-free survival; PFS, progression-free survival; OS, overall survival”.
We have checked that Figure 1 highlight the key proses and label the abbreviated enzymes and process within the legend and no modifications were reported.
- From line 63-69, it is in the wrong position I think this should be part of the figure legend.
Answer to question 4: The line 63-69 have been placed after the abstract, in the paragraph “Abbreviations”, Pages 1-2, lines 34-44.
- In vivo, vitro, ex vivo it should be written in italic please revise.
Answer to question 5: We have wrote in italic please revise: “in vivo” in Page 3, Line 84 and “ex-vivo” in Page 4, Lines 107-108.
- Is there a material section please list all chemical used and their suppliers ?
Answer to question 6: Thanks for your suggestion, we have added the following paragraph in “Material and methods”, in Page 5, Lines 169-174: “Patient's DNA is extracted from samples of 5 ml of peripheral blood, using the automatic QIA symphony system for the extraction of nucleic acid (Qiagen, Hilden, Germany) using the dedicated kits produced by the manufacturer of the instrument, then the latter is processed using a Next-Generation Sequencing platform Ion Chef/Ion S5 system (Thermo Fisher Scientific, Waltham, MA, USA) using the dedicated kits produced by the manufacturer of the instrument, according to the manufacturer's instructions”.
- Authors should provide an IRB reference for samples .
Answer to question 7: According to your suggestion we have added to “Materials and methods” the following paragraph in Page 6, Lines 209-213: “The study was conducted in accordance with the Declaration of Helsinki and all patients signed informed consent for scientific research purpose at the first oncological visit. Since the observational, non-intervational nature of this study, we just sent the local ethical committee (Comitato Etico Sapienza Università di Roma, Rome, Italy) a notification (normative ref. GU della Repubblica Italiana n.76 of 31 March 2008)”.
- Figure 2, same comment for the legend, and please use a clearer and better-quality font it is distorted
Answer to question 8: According to your suggestion we have replaced Figure 2 with an image of more quality, changing the colours of the boxes (blue) and words (white), thus making more visible the content, and we have added a legend for Figure 2 in Page 8, Lines 274-277, as follow reported: “150 colon cancer patients were assessed for eligibility. 120 out of 150 cases were excluded: 100 patients, assumed <5 concomitant drugs and 20 declined to participate. Colon cancer patients who were taking at least 5 drugs and candidate for chemotherapy were 43. Of these, 12 were excluded from analysis”.
- Could you please provide a better-quality Figure 2 very difficult to follow or use a yellow highlight with a black font?
Answer to question 9: According to your suggestion we have replaced Figure 2 with an image of more quality, changing the colours of the boxes (blue) and words (white), thus making more visible the content, in Page 8, Lines 244-273.
- If possible, replace the word “sex” in Table 1 with Gender and comprehensive legend.
Answer to question 10: According to your suggestion we have replaced the word “sex” in Table 1 with Gender, in Page 9, Line 284.
- Please align the G0-G2, G3G4 in table 2, and replace “sex” with gender.
Answer to question 11: According to your suggestion, we have aligned the G0-G2, G3G4 in table 2, and replace “sex” with gender, in Page 10, Line 302.
- Please include the conclusion section Information, Author Contributions, Conflict of Interest and other Ethics Statements following the journal formatting.
Answer to question 12: According to your suggestion, we have included Conclusion, Supplementary Materials, Author Contributions, Funding, Institutional Review Board Statement, Informed Consent Statement, Data Availability Statement and Conflict of Interest in Pages 12-13, Lines 402-420, as follow reported: “Conclusion . In conclusion, this study highlights the promising approach of the Drug-PIN® software in colon cancer patients and aims at enhancing the efforts on personalized medicine by tailoring therapies around the single patient, in order to prevent toxicities, increase compliance and therefore survival. Given these encouraging results, further studies are needed to confirm our data and better investigate the effect of Drug-PIN® in the new era of personalized medicine in cancer patients. Supplementary Materials: Supplementary Table 1: Panel of SNPs investigated. Author Contributions: Conceptualization, R.M. and R.A; methodology, R.M., R.A., P.L.M. and S.M.; validation, M.F., M.P., S.M. and R.M.; formal analysis, R.M., R.A., P.M, and P.L.M.; investigation, R.M., R.A., P.M., and P.L.M.; data curation, R.M., R.A., P.M., I.D. and A.G; writing—original draft preparation, R.M., R.A., P.M., and P.L.M.; writing—review and editing, R.M., R.A., P.M., and P.L.M.; supervision, M.P, M.F., and S.M.; project administration, R.M. All authors have read and agreed to the published version of the manuscript. Funding: this study has not benefited from any fund. Institutional Review Board Statement: The study was conducted in accordance with the Declaration of Helsinki and all patients signed informed consent for scientific research purpose at the first oncological visit. Since the observational, non-intervational nature of this study, we just sent the local ethical committee (Comitato Etico Sapienza Università di Roma, Rome, Italy) a notification (normative ref. GU della Repubblica Italiana n.76 of 31 March 2008). Informed Consent Statement: Informed consent was obtained from all subjects involved in the study. Data Availability Statement: Data is contained within the article or supplementary material. Conflicts of Interest: The authors declare no conflict of interest. The funders had no role in the design of the study; in the collection, analyses, or interpretation of data; in the writing of the manuscript, or in the decision to publish the results”.
- In the discussion could the authors add their reasoning of the section of DDI, DrugPin1, and DrugPin2 scores, and how do the findings corollate with previously published data.
Answer to question 13: According to your observation we have developed this argument by adding the following paragraph to the discussion in Page 11, Lines 324-335: “In fact, we defined for each patient the following crucial scores: DDI, DrugPin1 and DrugPin2. DDI is the score resulting from the interaction between medication taken by a patient. Basically, this is the “first step” of a complete evaluation. DrugPin1 score integrates medications with the genomic profile of each individual, and reveals a complex and unique drug-patient interaction that could lead to unexpected favorable or unfavorable outcomes. DrugPin 2 is the “last step”, adding chemotherapy to DrugPin 1 in order to highlight potentially dangerous interactions between our patient and the specific oncological treatment. Each of these steps must be monitored and considered individually, to better understand where potential relevant interactions might occur and provide a prompt solution. Most of the software do not integrate DDI with such wide range of single nucleotide polymorphisms and, moreover, they do not suggest which is the best available treatment for a particular patient relying on its individual profile. This is a completely new approach that takes personalized medicine at its best.
- Please include key references: 10.1038/s41397-019-0122-0; https://www.jpsmjournal.com/article/S0885-3924(19)30057-0/fulltext; https://doi.org/10.3389/fphar.2016.00071.
Answer to question 14: As you have recommended, we added key references: “https://www.jpsmjournal.com/article/S0885-3924(19)30057-0/fulltext and https://doi.org/10.3389/fphar.2016.00071 in introduction, in Page 2, Line 63 (references number 3-4); 10.1038/s41397-019-0122-0 in Page 3, Line 115 (reference number 21).
Furthermore we also have added the following reference in Page 2, Line 64: “Cabrera M, Finkelstein J. A Use Case to Support Precision Medicine for Frequently Hospitalized Older Adults with Polypharmacy. AMIA Jt Summits Transl Sci proceedings AMIA Jt Summits Transl Sci. 2016;2016:16-21”.
Round 2
Reviewer 1 Report
The revised manuscript improved the paper but still significant parts of it have not been revised in accord to the comments raised . Such comments are divided to those requiring extra experimental work and those that they can be addressed without experiments.
- G score power is a tool to compute statistical power analyses for many different t tests , F tests , Chi square test , Z tests and some exact tests.G power can also be used to compute effect of sample sizes and display graphically the results of power analysis .So please add how you calculated an appropriate statistical power for sample size . Mention this in methodology . This is standard practice .
- No survival curves , no univariate and multivariate analysis done .So can you please mention what standard metric you used to guage your results ?
- what you measured will be better alternative ? how it could be better that BNF ? Now-a-days we have BNF app. which everybody can have who is giving drugs to patients such doctors , nurses etc ..that contain all the information about drug interation , reaction , toxicities and also if patient is having specific genetic polypmorphism .Can you explain how your drug-pin techology is ''better''
Author Response
We are very pleased to receive your in-depth review of our manuscript. We appreciate your time and efforts in helping us improve our paper. We read your comments carefully and made corrections and revisions wherever possible. Our responses to each comment are below. We are looking forward to your further recommendations.
- G score power is a tool to compute statistical power analyses for many different t tests , F tests , Chi square test , Z tests and some exact tests. G power can also be used to compute effect of sample sizes and display graphically the results of power analysis .So please add how you calculated an appropriate statistical power for sample size. Mention this in methodology . This is standard practice.
Answer to question 1. Instead of G score power program we have used SPSS statistical software, version 24 (SPSS Inc. Chicago, Illinois, USA). We have reported statistical analysis in Page 5, Lines 189-196 and modified the paragraph as your suggestion “According to the aforementioned background, in order to provide an adequate sample size, assuming a 30% of moderate-severe toxicity with 5-FU based chemotherapy, a total of 81 patients will be enrolled (calculated with an error of 10% and confidence interval (CI) of 95%). However, a feasibility study analysis on the first cases enrolled was described in this publication. Associations between severe toxicities and clinicopathological characteristics, chemotherapy regimens and Drug-PIN® scores were assessed using the χ2–test and t-test to compare frequencies and means, respectively. A P value < 0.05 was considered as statistically significant. All the statistical analysis and sample size calculation were done with SPSS statistical software, version 24 (SPSS Inc. Chicago, Illinois, USA) .
- No survival curves , no univariate and multivariate analysis done .So can you please mention what standard metric you used to guage your results ?
Answer to question 2. Thanks to your observation, now we have reported the univariate analysis (HR and 95% CI) in Table 2, in Page 8, Line 258. We are conscious of limited sample size and that survival data are still immature. Indeed, we have reported in discussion section in Pages 9-10, Lines 311-315, the limit of our study by the text “Our study certainly presents some limitations to be taken into account: this is a brief report describing platform application with a limited number of patients; obtained results, although interesting, need to be validated with larger cohorts. Furthermore, there is a short-term follow-up, and data regarding disease-free (DFS), progression-free (PFS) and overall (OS) survival are still immature”.
- What you measured will be better alternative ? how it could be better that BNF ? Now-a-days we have BNF app. which everybody can have who is giving drugs to patients such doctors , nurses etc ..that contain all the information about drug interation , reaction , toxicities and also if patient is having specific genetic polypmorphism .Can you explain how your drug-pin techology is ''better''
Answer to question 3. The Drugpin system considers multiple aspects related to the patient (sex, age, voluptuous habits, gene polymorphisms) and his therapy as a whole, taking into consideration not only a one-to-one interaction between single drugs, but a global interaction of all the drugs assumed. Unlike other applications, Drugpin implements much more information and summarize it in a more intuitive, easy to use, way through a numerical score (defined in our study as: i) DDI is the score between those concomitant medications the patient is taking, ii) DrugPin1 is the score calculated before chemotherapy, including concomitant medications and SNPs profile; iii) DrugPin2 is the global score, including the DrugPin1 and chemo-drugs in the software analysis). In particular, it does not consider only the main known polymorphisms, for which in clinical practice there are already indications from international guidelines (for example DPYD or UGT1A1 to reduce the doses of chemotherapeutic drugs), but in our study, we were able to include in the analysis of Drug-Drug-Gene interaction for each patient, also genomic data of 110 SNPs.
We hope that we have done well
Best wishes
Martina Panebianco and colleagues